# Patient-Nurse Partnerships to Prevent Medication Errors: A Concept Development Using the Hybrid Method

**DOI:** 10.3390/ijerph19095378

**Published:** 2022-04-28

**Authors:** Hee-Ja Jeong, Eun-Young Park

**Affiliations:** 1Nursing Department, Shihwa Medical Center, Siheung-si 15034, Korea; jhj635175@gmail.com; 2College of Nursing, Gachon University, Incheon 21939, Korea

**Keywords:** partnership, patient–nurse relations, medication errors, proof of concept study

## Abstract

Medication safety is the most patient-centered aspect of nursing, and the medication process needs patients’ active participation to effectively prevent medication errors. The aim of this study was to develop the concept of a patient–nurse partnership for medication safety activities. The study design used the three-phase hybrid model for concept analysis: the theoretical phase, fieldwork phase, and final phase for integration. The results of a study define the concept of patient–nurse partnership for medication safety as “a fair cooperative relationship of mutual responsibility in which patients and nurses share information and communicate with each other through mutual trust.” Seven attributes were derived: mutual trust, mutual respect, mutual sharing, mutual communication, mutual responsibility, fair relationship, and mutual cooperation. The conclusion of the study of patient–nurse partnerships for medication safety was that it is necessary to ensure a balance in power between patient and nurse. This balance can be established through patient-centered nursing by implementing the active transfer of authority from nurses as professionals to patients.

## 1. Introduction

Medication safety activities are activities intended to correct and prevent possible medication errors [1]. The WHO [2] selected medication safety as the third patient safety challenge after infection prevention and surgical safety, and it has been making continuous efforts to reduce serious damage caused by medication errors [3]. Nevertheless, medication errors are experienced by 6–7% of inpatients worldwide [1]. In Korea, among medical errors, medication errors ranked second with a 31% rate of occurrence [4]. 

Medication errors have a direct adverse effect on the patient’s life and treatment and have a negative effect on the therapeutic effect by increasing the overall medical expenditures due to secondary costs, delaying treatment, incurring economic losses, and lowering trust in medical staff [5,6]. Medication errors by nurses occur during drug preparation, drug administration, and post-administration evaluation. Medication errors during drug administration and post-administration evaluation can be reduced with patient participation [7].

Therefore, prevention of accidents or errors that may occur during the medication process is an important medication safety activity. The best time to prevent medication errors in clinical settings is during the partnership, patient–nurse, and medication safety phase, when patient and nurse interact [8,9]. This period requires a patient–nurse partnership based on reciprocity.

A patient–nurse partnership enhances patient satisfaction, improves quality of life, and contributes to improving patient safety and healthcare quality through patients’ direct participation in decision-making for their own health management [10,11]. Moreover, it is an important factor in establishing a safe healthcare environment as a buffer to reduce patient safety incidents by facilitating communication [12]. Although the WHO has provided guidelines for the patient–nurse partnership for medication safety, it has not yet found applications in the clinical setting [1]. In the field of medication safety in particular, the only study on patient–nurse partnerships is an intervention based on a nurse–patient partnership program to prevent medication errors [13].

The concept of a patient–nurse partnership to prevent medication errors requires a clear definition because it can vary according to current trends and depends on the health care environment. The patient–nurse partnership has different meanings for different individuals, depending on their culture or the clinical setting [13,14], and according to disagreement among researchers about the definition of the concept. Thus, concept development can be used to resolve the ambiguities regarding the patient–nurse partnership and its use in clinical care. Conceptual analysis is required as it is useful for removing the ambiguity associated with a term and for studying a specific phenomenon [15]. 

As the prominent feature of the hybrid model for concept development lies in its stage of fieldwork [16], and because this study aims to show essential characteristics and an operational definition of the patient–nurse partnership in a new context, the hybrid model is the most appropriate method [17]. This study aims to analyze the concept of the patient–nurse partnership in medication safety.

## 2. Methods

### 2.1. Study Design

For conceptual analysis, the Schwartz-Barcott and Kim [15] hybrid model was selected as the most suitable approach toward concept analysis in this study. The hybrid model consists of three stages: theoretical, fieldwork, and analytical. This method has been used in nursing to investigate the patient–nurse partnership in the present context. This method is able to refine frequently-used concepts through inductive and deductive analysis.

### 2.2. Data Collection

#### 2.2.1. Theoretical Phase

The first part of the study was performed as a systematic review developed based on the stages proposed by the PRISMA guidelines [18]. The purpose of the theoretical phase is to lay the foundation for deep analysis, redefinition of the concept of the patient–nurse partnership, and identification of the antecedents and consequences of medication safety activities. For the literature search, the following keywords were selected: partnership, patient–nurse, and medication safety. We used the National Science Digital Library (NDSL), Research Information Sharing Service (RISS), DBpia, and Google as the Korean databases and CINAHL, PubMed, and Medline as international databases.

The search results yielded a total of 2915 international and 30 Korean papers for the documents published between 1978, which was the year when the WHO expressed the need for partnership and cooperation of all countries to ensure primary health care for all the people in the world, and 2019. After excluding overlapping and non-academic papers, 22 international papers and 1 Korean paper were finally selected for review in the nursing science field (Figure 1). 

#### 2.2.2. Fieldwork Phase

In the fieldwork phase, data were collected by conducting personal in-depth interviews with the selected cases judged to best reflect the concept of interest in light of the attributes identified in the theoretical phase. In addition, the conceptual attributes for the patient–nurse partnership for medication safety activities were determined.

The fieldwork was conducted after receiving approval from the Institutional Review Board of the Gachon University (1044396-201811-HR-193-01) and written consent from each participant for voluntary participation and recording. The study was conducted in accordance with the Declaration of Helsinki.

Participants were selected through purposeful sampling. Based on the literature review in the theoretical phase, the following inclusion criteria were applied: patients under constant medication with two or more hospitalization records; nurses with a clinical career of at least five years and who are capable of personalizing patient’s nursing needs, managing situations, and solving problems in special situations. Five patients and six nurses participated. Although 10 patients and 10 nurses were recruited by the recommendations of healthcare managers and using the snowball sampling method, five patients were unable to participate due to discharge and transfer and four nurses were excluded due to work scheduling issues. A total of 11 participants met the guidelines specified by Schwartz-Barcott and Kim [14]. The participants’ general characteristics are presented in Table 1.

##### Data Collection and Analysis

Data were collected from November 2018 to November 2019 through personal interviews, and each interview was conducted at a time and place chosen by the participant. It was conducted in the places preferred by the participants, such as the participant’s house, coffee shop, or conference room. Prior to the start of the interview, each participant was provided an explanation of the purpose of the study, interview contents, and expected interview duration, and that the interview would be recorded. Each participant was interviewed once or twice, and each session lasted between 30 and 120 min. When new information could not be retrieved from the interview, it was determined that the data had reached saturation and the data collection was terminated. The researcher transcribed the recording of each interview session. If additional questions were raised in the post-transcriptional analysis process, a second interview was conducted either face-to-face or by phone or email. An additional interview was conducted with nine participants.

The interview was conducted by a researcher who is a nurse who worked as a quality improvement specialist and used semi-structured questions based on the attributes derived in the theoretical phase and consisted of both specific and common questions. Common questions asked to both patients and nurses were: What do you think about safe medication? Tell me your role in the medication process. Tell us how to build the patient–nurse relationship for safe medication. The question specific to patients was: If there was a situation of partnership for your safety during injection or oral medication administration, please tell me about your feelings or experience at that time. The question specific to nurses was: If there was a situation in which you needed partnership with a patient in relation to medication care, please tell me about your experience or feelings at that time.

Among the qualitative content analysis methods suggested by Elo and Kyngäs [19], we used the inductive content analysis method. First, while repeatedly reading the transcribed material, we marked meaningful statements related to the patient–nurse partnership for medication safety, and then we derived attributes. The derived attributes were categorized into sub-attributes under generic terms, and these sub-attributes were grouped together and abstracted into an overarching attribute. In this process, if the name of the attribute identified in the theoretical phase was used in the same sense as in the fieldwork phase, it was thus named, and the attribute newly derived in the fieldwork phase was added to the attribute according to its meaning.

##### Ensuring Rigor

To ensure the rigor of this study, we applied the criteria proposed by Sandelowski [20]: reliability, fittingness, auditability, and confirmability. The data from this study reveals the actual experiences of the patients and nurses. The interviews were recorded and transcribed verbatim, and we tried to ensure the accuracy of the transcriptions by comparing them with the recordings. We also identified the adequacy of citations to support themes derived from the data. The participants’ narratives were described and interpreted based on the derived themes. We read the manuscript multiples times and clearly described the analytical procedures for deriving the initial codes, sub-themes, and themes. Two researchers were involved in all data analysis processes. We also tried to maintain neutrality throughout the research process. In addition, an attempt was made to increase the reliability of the research results through feedback between researchers and member checks, including two nurses and one patient participant.

### 2.3. Final Analysis Phase

The final analysis phase is a process of integrating the results of the theoretical and fieldwork phases. Two questions were addressed in this phase: How consistent are the results of the literature review and fieldwork regarding the conceptual attributes and definitions of the patient–nurse partnership for medication safety? How do the attributes and definition of this concept find applications in the nursing environment? Based on these questions, the results of the theoretical and fieldwork phases were analyzed in an integrated manner, and the conceptual attributes of the patient–nurse partnership for medication safety were reorganized and defined.

## 3. Results

### 3.1. Theoretical Phase 

#### 3.1.1. Definition of Partnership in Other Academic Disciplines

The dictionary definition of a partnership is “a cooperative relationship in which individuals or groups contribute resources, share profits and losses, and take responsibility to achieve specific goals” [21]. Partnerships are prominent in a variety of fields, particularly economics, sociology, and health care. A partnership in the political and social field refers to “a fair and cooperative relationship in which partners share professional information based on mutual understanding and respect to achieve their goals” [22]. Partnership in the health care sector refers to “patients and health care providers are in a therapeutic partnership in which they fulfill their mutual roles and responsibilities” [23].

#### 3.1.2. Partnership in Nursing 

As a result of analyzing a total of 23 nursing papers, we defined the concept of partnership in nursing as “a fair cooperative relationship in which patient and nurse communicate with each other based on mutual trust and respect to achieve the goal of health recovery, playing their respective roles and assuming their respective responsibilities.” Patients and nurses are required to sympathize with each other, listen to each other, have trust in each other, and be considerate of each other. They also need to have mutual respect for their rights and patient autonomy based on patient-centeredness. Additionally, the patient’s health information and nurse’s professional knowledge should be mutually shared to benefit the patient’s own decision-making, and mutual communication is required for efficient bidirectional communication. A fair relationship of mutual equity between partners who understand a common goal is maintained through cooperation between patient and nurse, enabled by the patient’s active participation.

#### 3.1.3. Conceptual Attributes and Definition of Partnership in the Theoretical Phase

The attributes of the concept of the patient–nurse partnership in patient safety activities analyzed through the literature review are mutual trust, mutual respect, mutual sharing, mutual communication, mutual cooperation, mutual responsibility, and a fair relationship. Among the derived attributes, those common to other disciplines are mutual trust, mutual respect, mutual sharing, and communication, and those specific to nursing are mutual responsibility and fair relationship. Table 2 outlines the attributes (and their respective meanings) of the patient–nurse partnership derived in the theoretical phase.

### 3.2. Fieldwork Phase

Five themes of the patient–nurse partnership for medication safety were derived in the fieldwork phase: nurse’s role, patient’s role, trust in experts, collaborative relationship, and power imbalance. 

#### 3.2.1. Nurse’s Role

##### Provision of Accurate Medication Information

Provision of accurate medication information means providing information on the medication being administered to the patient. It is a core role perceived by all nurse participants in the medication safety experience. Provision of accurate medication information to patients equips them with proper knowledge of the medication and enables them to prevent medication errors and correctly respond to incidents of medication errors. It is also an opportunity to elicit questions from the patients. 

The same attribute was also derived regarding the role of the nurse, as expressed by the patient-participants. The patients felt they were not sufficiently informed about medication, although they had plenty of questions about the medication they were taking to treat their illnesses, the period and method of taking it, and precautions, and expected the nurse to provide them with concrete medication information.


*The medication information provided by the nurse may not be what the patient wants. Nurses attach importance to the name of the drug, but as a patient, it may be of more interest to know why I have to take that medication, what it does when it enters my body, what side effects it may have, and what to eat or not to eat (Nurse participant C).*



*It’s frustrating to know just the drug name I take. They don’t explain… It’s good to know what it is good for and why I have to take it. You don’t have to do it every day… I don’t even want them to tell me every time… It would be nice to be informed when I take a new drug and when they change drugs... (Patient participant C).*


##### Adherence to Medication Principles and Patient Checks

A nurse’s adherence to medication principles means the nurse’s duty toward medication safety. The nurse participants noted that they checked the dispensed medication against the doctor’s prescription before administration and confirmed the appropriateness of administration by checking the patient’s medication information and health status, such as allergy history and ongoing medications.


*Listening to a patient’s story to the end is important because there may be allergic reactions that the patient has not been able to tell. It is also important to ask questions to check the patient’s condition and whether the medication agrees with the patient’s system (Nurse participant C).*


##### Sharing Medication Errors with the Patient

The nurse participants stated that sharing medication errors with the patient is an attempt to prevent their recurrence and to respect the patient. However, because medication errors lead to complaints, nurses do not always report them, especially when no harm was done to the patient. For medication safety, it is necessary to change the patient–nurse trust relationship and perception of medication errors. The sharing of responsibilities between patient and nurse through medication error disclosure can contribute to enhancing medication management safety. 


*Sometimes we report medication errors to the attending physician and only check for adverse effects of medication. (…) If a patient learns about the medication error incident, it can result in a serious complaint. So, unless serious harm is done to the patient, I often keep it to myself without reporting it (Nurse participant D).*


#### 3.2.2. Patient’s Role

##### Checking

This attribute refers to the process of getting the patient’s name and medication checked by the nurse at the time of medication administration. Oftentimes, however, it is passive rather than voluntary confirmation. In particular, the patient participants who had experienced receiving other patients’ medications in error said that they were more active in checking the medication for their own safety. 


*My frequent hospitalizations have made me a veteran when it comes to medication. I know what I’m taking, and I know that I was given others’ medications several times, and strange drugs were mixed... At first, unfamiliar with the hospital environment, I was quite sensitive and I complained to the nurse in a reproachful tone, like “It’s hard enough to get my own body going, and now I have to take care of my medications even in a hospital.” But I have a changed mind after it happened repeatedly. It’s my health, so I’ll take care of my own medications… (Patient participant E).*


##### Asking

Nurse participants mentioned that when a patient asks a question about the prescribed medication, they faithfully provide the medication information. Patient participants called for a change in the mindset of healthcare personnel who perceive patients’ questions about the medication as an expression of mistrust toward the nurse. Patients also mentioned the need for a hospital culture that allows patients to ask questions at any time. 


*Patients ask about their medications, like “Why is this here? What is this for?” Then I think of the diagnosis and answer, “This is for…” or explain again “This was added as a gastric ulcer treatment after endoscopy, so it is better to take it before meals” (Nurse participant A).*



*In our culture, even though I ask in a friendly and polite way, they are annoyed and think I do not trust them (Patient participant A).*


#### 3.2.3. Trust in Experts

##### Unconditional Trust

In Korea’s healthcare culture, a patient takes any medication administered by a nurse in blind trust as long as his or her name is on the pill pouch. In addition, patients usually check only the time to take the medication, so there is no way of being informed of medication errors unless fatal side effects occur. Patients have no choice but to trust nurses. 


*Once my name is written on the pill pouch, I just trust it 100% and take it (…) How many people would ever open the pill pouch and look at the color and shape of each pill? Especially patients who are hospitalized because they are sick… (Patient participant A).*



*A patient X, who is often hospitalized here trusts nurses blindly. There have been medication errors that could have been avoided with some care. (…) Not that it is bad to trust nurses too much... It’s no use to say “Please check your name” when taking medicine, because of his unconditional trust in nurses… (Nurse participant C).*


##### Fair Relationship Based on Trust

Nurse participants noted that “fair trust” between patients and nurses was necessary for medication safety. The nurse should provide medication information to the patient, and the patient should share their medication history with the nurse, thus establishing a fair relationship for medication safety. On the other hand, the patient participants requested that the nurses take the initiative in starting a fair relationship between patient and nurse while respecting the patients because patients have an inferior status in healthcare facilities. 


*I think patients and nurses should be fair to each other. It does not mean equality, given that each patient’s situation is different. In order to customize a medication for each patient, it is necessary to adopt a personalized approach, checking the appropriateness of the prescription considering the patient’s condition, including allergy history and interaction with the current medication (Nurse participant F).*



*Nurses respect patients’ opinions, and patients comply with nurses’ instructions. Above all, I think patients and nurses should trust each other and be considerate of each other. Only then can they respect each other (Patient participant E).*


#### 3.2.4. Conditions for a Cooperative Relationship

##### Nurse’s Professional Interest and Attention 

Nurses emphasized the importance of professional attention to medication safety during the medication administration process. They use their own methods to share the medication process with the patient. In this context, they approach the patient, collect the necessary information, and engage the patient in the medication process, thus practicing safe medication. 


*I even position the injection ticket toward the patient’s direction as if to demonstrate “Take a look. I’m doing it alright.” When you show the medication information to patients, they take interest, saying, “Oh, is this my medication card?” (Nurse participant A).*



*Above all, I think the nurse should give the patient medication information. How can the patient ask about medication if they have no information about it? (Nurse participant B).*


##### Active Patient Participation

The nurse participants found it necessary for patients to stop taking their medication in case of doubt and to check it with the nurse in charge. When the patient asks a question, the nurse will check the medication and provide the medication information again. It is hence important for medication safety that patients are actively involved in the medication process. 


*In case of a name error, they have only to look at the pill pouch. If patients check for their names on the pill pouch, medication errors from taking false medications would not happen… They just take anything in the pouch without looking at the name on it (Nurse participant C).*



*If any patient would like to know what they are taking, I feel a little bit disturbed, but I explain once again… (Nurse participant A).*


#### 3.2.5. Characteristics of the Patient–Nurse Relationship

##### Uneven Contractual Relationship

The patient participants noted that the patient–nurse relationship should be on an equal footing because the nurse provides nursing care and the patient pays for the nursing service through a sort of contract. Providing medication information is, therefore, the nurse’s duty when administering it, and it is the patient’s right to be given an explanation.

One of the patient participants said that even if he suffers a physical injury in the event of a medication error, he feels like a sinner just by being there and relying on the nurses’ care for his health recovery. Some of the nurse participants also expressed the view that the patient gown seems to put patients in an inferior position. However, other nurse participants had the opposite experience of being exposed to patients’ excessive demands and complaints and felt that patients had the upper hand. This attribute was common among patients and nurse participants. While both groups were of the view that the patient–nurse relationship should be a contractual relationship established on an equal footing, each group believed that the other group had the upper hand. 


*It feels as if my life is placed in the nurses’ hands. But what could I do… I am sick and I need their help... (Patient participant C).*



*I have only changed into the patient gown and that puts me in a totally different position… I have put my health into their hands. The hospital encourages us to report complaints or dissatisfactions, but it is difficult from the position of a patient (Nurse participant B).
*


##### Patient’s Right to Respectful Care 

Nurse participants found that patients are hesitant about actively participating in their medication process and asserting their rights. Of course, patient questions and requests can act as additional burdens for nurses because their workload will increase and they will have to gain further professional knowledge. However, because a safer healthcare environment can be ensured only when they work with patients, the nurses found it necessary for patients to strengthen their claim on patient safety rights.


*It would be really helpful if patients take more interest in their medications and simply ask “What is this? Why is this in my pill pouch?” … It’s their own right, but, unfortunately, they don’t do it. (…) Patient participation will certainly reduce medication errors. Nurses will be busier and annoyed, but it’s worth the effort (Nurse participant A).*



*Patients themselves should take interest in the medication administration and be able to check with the nurse what medications they need to take and how to administer them. At the moment, however, it is urgent to raise their awareness first (Nurse participant E).*


#### 3.2.6. Conceptual Attributes and Definition of Partnership in the Fieldwork Phase

As the themes similar to attributes were derived in the theoretical phase, those pertaining to the roles of patients and nurses and the patient–nurse relationship were additionally derived. The attributes of the concept of the patient–nurse partnership in patient safety activities analyzed through the voices of patients and nurses are mutual responsibility (including the roles of nurse and patient), mutual trust, mutual cooperation, and mutual respect. However, mutual sharing, mutual communication, and a fair relationship were premised on the overall theme. Table 3 outlines the attributes of the patient–nurse partnership for drug safety derived in the fieldwork phase.

### 3.3. Final Analysis Phase

In the final analysis stage, concepts and attributes that fit the purpose of the study were searched for while continuously comparing and analyzing the attributes of partnerships derived from the theoretical and fieldwork stages. As a result, the same properties derived in the theoretical and field research stages were maintained without modification. If the attributes had different definitions but the same meanings, they were integrated together into an overarching attribute. Among the attributes derived from only one of the two phases, those containing mutuality, the basic attribute of partnership, or basic principles for medication safety, were selected as an attribute of this study.

The seven attributes derived in the final analysis phase can be divided into three categories: (1) Attributes common to both theoretical and fieldwork phases: mutual cooperation and mutual responsibility; (2) Attributes integrated into an overarching attribute: mutual trust and fair relationship. Mutual trust represents the trust and consideration formed between patient and nurse by listening to and sympathizing with each other. It was derived in the fieldwork phase as trust in experts in the sense of fair trust rather than the patient’s unconditional trust in nurses. Because it implies relational balance and respect for patient’s rights, mutual trust was integrated into “fair relationship,” representing a patient-centered, bidirectional, and symmetrical therapeutic relationship derived in the theoretical phase; (3) Attributes embracing mutuality: mutual respect, mutual sharing, and mutual communication (Table 4). 

## 4. Discussion

The main point at which a nurse’s dosing error occurs is during contact with the patient [7]. Nursing is not provided unilaterally by the nurse but holds a patient’s health care journey together [24]. The medication process is no exception. The patient–nurse partnership based on reciprocity derived from the medication process is an important point in medication error prevention and safe medication administration. This study conceptualized active and voluntary participation in the medication process as a patient–nurse partnership and explored its attributes.

The patient–nurse partnership for medication safety derived from this study was defined as “a fair, cooperative relationship of mutual responsibility in which patients and nurses share information and communicate with each other through mutual trust and respect.” Seven attributes were derived: mutual trust, mutual respect, mutual sharing, mutual communication, mutual responsibility, fair relationship, and mutual cooperation. These results were similar with nurses involved in the process of medication administration as an opportunity to engage patients in a conversation about medications and safety [13].

The attributes of mutual trust, mutual respect, mutual sharing, and mutual communication were verified as the attributes of the patient–nurse partnership in medication safety in the same sense used in previous studies [25,26,27]. They are attributes inherent in partnership and can be regarded as necessary components of all circumstances and relationships where reciprocity is assumed. Among the results of this study, the partnership-related attributes and circumstances that were not derived from previous studies were: a safe medication care environment, a fair relationship, and mutual responsibility. 

The attribute “fair relationship between patient and nurse” refers to a patient-centered, bidirectional, and symmetrical therapeutic relationship in the field of medicine [26,28,29]. However, in a patient–nurse relationship in actual healthcare settings, patients cannot choose their partners. That is, the imbalance of power starts when the relationship is formed. As suggested by Falk, Schandl, Frank [30], and Sewell [31], while the clinical setting calls for patient-centeredness, it is too early to broach the matter of partnership because of the current power imbalance. Therefore, it is natural that patient-centered nursing, which transfers the nurse’s authority to the patient, is emphasized as a process that establishes the balance of power. Winslow [27] also emphasized the need to allocate nursing resources on the basis of fairness to improve quality of life and patient satisfaction in clinical practice. In addition, previous studies [27,31] also pointed out that effective communication skills and delegation ability based on patient–nurse interaction are necessary. In conclusion, in order to maintain a fair relationship that emphasizes patient-centeredness, emphasis should be placed on the mutual responsibility of patients and nurses.

The attribute “mutual responsibility” refers to the adherence to mutual obligations between patients and nurses. In a partnership, mutual responsibility is an essential condition for maintaining it [8,23]. However, given that a patient–nurse partnership begins in an assigned relationship without the process of partner selection, its characteristics are different from those of a partnership in other disciplines, where partners are arbitrarily selected depending on the goal [22]. The patient’s passive role can lead to medication errors, even when a medication safety system is in place [1]. Therefore, in order to ensure patient safety, equal emphasis should be placed on the responsibilities of both patients and nurses.

The patient’s responsibility is to recognize their own medications and the corresponding medication information, as suggested in previous studies [30,31,32], provide the nurse with a history of drug allergies or current medication information, and if necessary, seek answers to their questions. Another aspect of partnership formation was added in this study: a change in the mindset of healthcare personnel; the patient’s question is a basic task for patient safety, not intended as monitoring, and represents the patient’s active assumption of responsibility.

The results of this study highlight the importance of establishing a proper balance of power between patients and nurses as a prerequisite for establishing a proper partnership for medication safety in healthcare settings. The balance of power between patient and nurse can be established through patient-centered nursing through the active transfer of authority from professional nurses to patients. To achieve this, it is necessary to support the work, human resources, and education systems of healthcare facilities and nursing organizations at the government level, while considering various clinical settings. Therefore, in order to establish a patient-centered healthcare environment aimed at achieving patient safety, healthcare personnel and systems should strive to foster a culture that recommends and encourages patient responsibility and participation.

The concept of the patient–nurse partnership for medication safety activities is expected to find applications in various patient safety activities aimed at establishing a safe medical environment. To further improve the sophistication and generalizability of the concept of the patient–nurse-partnership for medication safety, we suggest in-depth, duplicated research on patients with various diseases and medication experiences and nurses from various fields. We also suggest follow-up studies to develop a patient–nurse partnership measurement scale with proven validity and reliability based on the concept of the patient–nurse partnership for medication safety. A study on medication preparation activities from the nurse’s perspective to prevent medication errors in the future is proposed.

The limitations of this study are that the results cannot be generalized to all patient–nurse partnership clinical nursing settings. This is because the nurses’ working conditions, such as staffing and a safe medication care environment, and the individuals’ characteristics, as well as the patients’ disease-specific characteristics, are not representative of all clinical nursing settings. Another limitation is that it is limited to partnerships with patients who can communicate verbally. It cannot be applied to all patients because it does not include the various conditions of patients who are the subjects of drug safety activities. In addition, it does not contribute to the prevention of medication errors that may occur in the medication preparation stage rather than at the contact point with the patient.

## 5. Conclusions

This study establishes the role of patients and nurses in medication safety activities by presenting a clear concept of the patient–nurse partnership for medication safety for nursing phenomena that were vaguely used in clinical nursing. The patient–nurse partnership in medication activity includes the property of a cooperative relationship in information sharing and goal achievement of partnerships in other academic fields, but there is a difference in fairness. In addition, although it includes the attributes of therapeutic partnerships that perform mutual roles and responsibilities of partnerships in the medical field, there is a difference in emphasizing the attributes of trust.

In addition, we suggest that the concept of patient–nurse partnership for medication safety can provide the basis and protocol for educational materials for nursing activities related to patient safety in medication.

## Figures and Tables

**Figure 1 ijerph-19-05378-f001:**
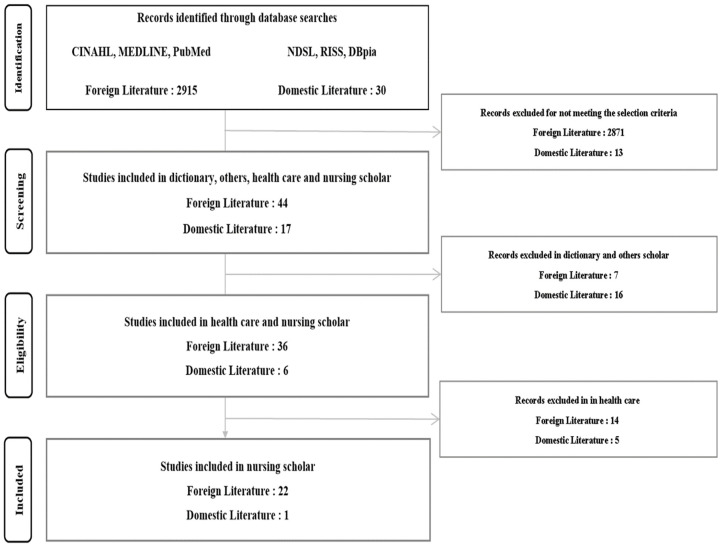
Flowchart of decisions made for selection, critical appraisal, and data extraction for the theoretical phase.

**Table 1 ijerph-19-05378-t001:** General Characteristics of Participants.

Participant		Age	Gender	Job	Number of Admissions	Medication Error Experience
Pt	A	54	M	Yes	2	0
	B	64	F	No	2	1
	C	51	F	No	4	1
	D	65	F	Yes	3	1
	E	64	F	Yes	10	3
Nr		Age	Gender	Department	Years of career	Years of career of patients safety
	A	36	F	Ward	14	
	B	38	F	Ward	11	
	C	43	F	Ward	9	
	D	36	F	Q.I Team	14	2
	E	40	F	P.I Team	17	4
	F	51	F	P.I Team	28	4

PI = Performance Improvement; QI = Qualitative Improvement; Pt = Patient; Nr = Nurse.

**Table 2 ijerph-19-05378-t002:** Attributes of Patient–Nurse Partnership in Theoretical Stage.

Attributes	Explanation	Reference
Mutual cooperation	Mutual cooperation between patients and nurses through active involvement of patients.	Craske et al. (2019) Doss et al. (2011) Hopwood et al. (2018) McTier et al. (2015) Phipps et al. (2018) Vasey et al. (2019) Vonnes & Wolf (2017)
Fair relationship	A bidirectional and symmetrical therapeutic relationship for mutual understanding and resource sharing for health recovery, sharing gains and losses, and negotiating roles.	American Heritage Dictionary (2006)
Mutual Responsibility	Patient’s roles: Asking questions to obtain information on his/her own conditions and to obtain help to better understand the information provided.	Hammoudi et al. (2018) Nabavi et al. (2017)
	Nurse’s roles: Sharing professional knowledge and skills with patients. Showing sensitivity to access the patient according to his/her condition and state Adherence to nursing principles.	
Mutual communication	Simple, clear, and active communication for effective communication regarding the goals and roles for health recovery.	Car et al. (2017) Hopwood et al. (2018) Phipps et al. (2018) Roach & Hooke (2019) Son et al. (2018)
Mutual sharing	Sharing patient’s health information by sharing the goals for health recovery. Sharing professional nursing knowledge and skills.	Craske et al. (2019) Doss et al. (2011) Hammoudi et al. (2018) Hopwood et al. (2018) Son et al. (2018)
Mutual respect	Respecting the patient’s demand and autonomy based on patient-centeredness Sharing decision-making. Empowering each other and recognizing each other’s abilities.	Doss et al. (2011) Eassey et al. (2019) Hopwood et al. (2018)
Mutual trust	Trust and consideration through mutual listening and empathy between patients and nurses.	Car et al. (2017) Hammoudi et al. (2018) Phipps et al. (2018)

**Table 3 ijerph-19-05378-t003:** Component Attributes of Patient–Nurse Partnership in Field Work Stage.

Attributes	Themes	Sub-Themes
Mutual responsibility	Nurse’s roles	Provision of accurate medication information
		Adherence to medication principles and patient check
		Sharing medication errors with the patient
	Patient’s roles	Checking
		Asking
Mutual trust	Trust in experts	Unconditional trust
		Fair relationship based on trust
Mutual cooperation	Conditions for cooperative relationship	Nurse’s professional interest and attention
		Active patient participation
Mutual respect	Characteristics of patient–nurse relationship	Uneven contractual relationship
		Patient’s right to respectful care

**Table 4 ijerph-19-05378-t004:** Components of Final Confirmed Patient–Nurse Partnership.

Attributes	Explanation Related to Partnership	Situation in Context
Mutual trust	Trust and consideration between patients and nurses through mutual listening and empathy	Medication information considering the patient’s state
		Attentive listening to patient’s needs
		Consideration of nurses to maintain medication care
		Patient’s adherence to nurse’s requests
Mutual respect	Respect for the rights and autonomy of patients and the rights granted to patients and nurses	Respect for patient rights
		Patient respect for nurse’s authority
		Positive acceptance of patient questions
		Patient’s understanding check after providing medication information
Mutual sharing		Patients sharing their medication information with nurses
		Sharing professional knowledge/skills on medications with patients
Mutual communication	Efficient communication for rational decision-making	Communication regarding medication information
		Adequate communication with patients refusing medications
Mutual responsibility	Patient’s role	Transmitting the patient’s medication information to the nurse
		The patient’s perception of medication information
		Checking the name of the medications delivered
		Asking and clarifying medication-related doubts
	Nurse’s role	Confirmation of the prescribed and dispensed medications
		Checking the patients medication information and status
		Access to medications according to the patient’s condition
		Compliance with medication administration principles
		Provision of medication and administration information
Fair relationship	A patient-centered, bidirectional and symmetrical therapeutic relationship of sharing and understanding goals, sharing gains and losses, providing resources, and negotiating roles.	Sharing and understanding the patient’s health recovery goals
		Sharing the results and responsibilities for medication errors
		Patient-centered work progress and problem-solving
		Guaranteeing the patient’s right to ask
		Creating the environment for the patient to ask
Mutual cooperation	Active and committed participation of patients and nurses	Cooperation with the nurse during nursing care
		Immediate medication cessation if other medication is provided

## Data Availability

The data that support the findings of this study are available from the corresponding author, upon reasonable request.

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
