# Peer review of "Patient-Nurse Partnerships to Prevent Medication Errors: A Concept Development Using the Hybrid Method"

_ijerph, 2022, doi:10.3390/ijerph19095378_

Round 1

Reviewer 1 Report

Thank you for the possibility to review this article.

I consider the topic and the results to be very important and crucial for nursing practice and nursing education.

The strong point of this work is, in my opinion, a combination of research methods and the specific characteristic of the patient-nurse partnership. The weak point of this work is the application of research criteria according to Sandelowski from the year 1986 and the absence of PRISMA diagram with evidence of the article selection. On the other side, I understand that the range of the text is limited. 

I recommend the authors to add the type of the study to the title of the article and to correct the title of Chapter 3: Results instead of the Methods.  

Author Response

1. The strong point of this work is, in my opinion, a combination of research methods and the specific characteristic of the patient-nurse partnership. The weak point of this work is the application of research criteria according to Sandelowski from the year 1986 and the absence of PRISMA diagram with evidence of the article selection. On the other side, I understand that the range of the text is limited. 

Answer: We add the PRISMA diagram with evidence of the article selection in <Figure 1>

2. I recommend the authors to add the type of the study to the title of the article and to correct the title of Chapter 3: Results instead of the Methods.  

Answer: Thank you. I corrected the term of Chapter 3 to Results, and the title of study to

Concept Analysis of Patient-Nurse Partnerships to Prevent Medication Errors: A Hybrid Model.

Reviewer 2 Report

The manuscript Ijerph-1680955, entitled "Concept Analysis of Patient-Nurse Partnerships to Prevent Medication Errors", analyses the concept of patient-nurse partnerships within the context of safe medication management for patients. 

The introduction and the theoretical framework precisely delimit the state of the question and the content analysis of the data is an appropriate method for the achievement of the objective. The wording of the objective, the presentation of the results, the discussion and the elaboration of the conclusions is coherent and presented in a way that is easy to follow. The wording and order of presentation is appropriate and makes the manuscript easy to read.  

However, there are some methodological quality issues that are not specified in the manuscript and which it is recommended to include, as well as some interviewer-specific considerations.  

For these reasons, a "minor revision" of the present version is recommended before publication.

Specific remarks:

0- Abstract: adequate and structured.  

1- Introduction:

It is brief and adequate, emphasising and defining the essential concepts of the study. The methodological orientation and theory underpinning the study is accurately described. 

However, in the description and justification of the hybrid model used (from Schwartz-Barcott & Kim) the information contained in the following lines is repetitive (L.56-L71). This same information is repeated again in the first paragraph of "material and methods" (L73-75). It is suggested to rewrite this fragment avoiding the repetition of information, eliminating the repeated content and selecting the place it has to occupy in the manuscript. 

In addressing the state of the art, a previous study is referenced, but the main findings obtained are NOT indicated. 

  1. Methods:

2.1.Data collection: 

It is recommended to include the characteristics of the interviewer: gender, possible biases and interests in the research. 

In the theoretical phase, the Prisma statement is stated, but the inclusion and exclusion criteria used and the typology of the 23 articles finally included are not specifically addressed. 

In the fieldwork phase, the sample design is sufficiently explicit, both in the selection of participants, the sample design, the method of approach and the reasons for non-participation. 

However, in the data collection, it is recommended to include the place where the data were collected, to clarify the presence or not during the interviews of "non-participants", and whether or not data saturation was discussed. Finally, it is relevant to indicate whether the transcripts were returned to the participants for possible corrections as well as the method used to obtain participants' feedback on the findings. 

  1. Results: The heading of this section is wrong, it refers to Results, not Methods. Correction. L163

The development of the results is adequate, addressing in an orderly manner the results of each phase of the research. The results are supported in a classified manner in table 2 and table 3. To facilitate reading and comprehension, it is recommended that the position currently occupied by the tables in the manuscript be modified, moving them to a position before the text in which their content is detailed. This suggested earlier position would allow a global visualisation of the attributes and subsequently facilitate the monitoring of the results obtained.  

  1. Discussion

It is coherent and is written in an orderly and integrated manner.

In the limitations section, no limitations specific to the study are included, the only limitation being the one described for the qualitative approach in general terms: the impossibility of generalising the findings. 

The express inclusion of the limitations detected in the design or in the development of the study is necessary in the manuscript. 

  1. Conclusion

Consider including the information contained in L452-459 in the discussion, not in the conclusions section. The latter should give a clear answer to the objective, not refer to possible future lines of research. 

L-462: it is recommended to replace the verb "believe" with a very subjective character by another verb that indicates the consideration of a recommendation based on the findings.

Author Response

1- Introduction:

It is brief and adequate, emphasising and defining the essential concepts of the study. The methodological orientation and theory underpinning the study is accurately described. 

However, in the description and justification of the hybrid model used (from Schwartz-Barcott & Kim) the information contained in the following lines is repetitive (L.56-L71). This same information is repeated again in the first paragraph of "material and methods" (L73-75). It is suggested to rewrite this fragment avoiding the repetition of information, eliminating the repeated content and selecting the place it has to occupy in the manuscript. 

In addressing the state of the art, a previous study is referenced, but the main findings obtained are NOT indicated. 

 Answer: In the introduction, the necessity of conceptual analysis was described, and detailed contents were described in the study design.

2. Methods:

2.1.Data collection: 

It is recommended to include the characteristics of the interviewer: gender, possible biases and interests in the research. 

Answer: We add the sentence on ‘Data collection and analysis’: The interview conducted by researcher who is a nurse who worked as quality improvement specialist

In the theoretical phase, the Prisma statement is stated, but the inclusion and exclusion criteria used and the typology of the 23 articles finally included are not specifically addressed. 

Answer: We add the PRISMA diagram with evidence of the article selection in <Figure 1>

In the fieldwork phase, the sample design is sufficiently explicit, both in the selection of participants, the sample design, the method of approach and the reasons for non-participation. 

However, in the data collection, it is recommended to include the place where the data were collected, to clarify the presence or not during the interviews of "non-participants", and whether or not data saturation was discussed.

Answer: We add the sentence on ‘Data collection and analysis’: It was conducted in the place preferred by the participants, such as the participant's house, coffee shop, and conference room. And   When new information could not be retrieved from the interview, it was judged that the data had reached saturation and data collection was terminated.

The information of participants who declined to be interviewed was describe as, five patients were unable to participate due to discharge and transfer and four nurses were excluded due to work scheduling issues(L108-110).

Finally, it is relevant to indicate whether the transcripts were returned to the participants for possible corrections as well as the method used to obtain participants' feedback on the findings. 

We add the sentence on final part of ‘Data collection and analysis’: In addition, it was attempted to increase the reliability of the research results through feedback between researchers and member check including 2 nurses and 1 patient participants.

3. Results: The heading of this section is wrong, it refers to Results, not Methods. Correction. L163

The development of the results is adequate, addressing in an orderly manner the results of each phase of the research. The results are supported in a classified manner in table 2 and table 3. To facilitate reading and comprehension, it is recommended that the position currently occupied by the tables in the manuscript be modified, moving them to a position before the text in which their content is detailed. This suggested earlier position would allow a global visualisation of the attributes and subsequently facilitate the monitoring of the results obtained.  

 Answer: Thank you. We correct the term to Results.

4. Discussion

It is coherent and is written in an orderly and integrated manner.

In the limitations section, no limitations specific to the study are included, the only limitation being the one described for the qualitative approach in general terms: the impossibility of generalising the findings. 

The express inclusion of the limitations detected in the design or in the development of the study is necessary in the manuscript. 

 Answer: We add the limitations.; Another limitation is that it is limited to partnerships with patients who can communicate verbally. It cannot be applied to all patients because it does not include the various conditions of patients who are subjects of drug safety activities. In addition, it does not contribute to the prevention of medication errors that may occur in the medication preparation stage rather than the contact point with the patient. A study on medication preparation activities from the nurse's perspective to prevent medication errors in the future is proposed

5. Conclusion

Consider including the information contained in L452-459 in the discussion, not in the conclusions section. The latter should give a clear answer to the objective, not refer to possible future lines of research. 

Answer: According to your opinion, rewrote the conclusion, add the below,

The nurse-patient partnership in medication activity includes the property of a cooperative relationship in information sharing and goal achievement of partnerships in other academic fields, but there is a difference in fairness. In addition, although it includes the attributes of therapeutic partnerships that perform mutual roles and responsibilities of partnerships in the medical field, there is a difference in emphasizing the attributes of trust.

L-462: it is recommended to replace the verb "believe" with a very subjective character by another verb that indicates the consideration of a recommendation based on the findings.

Answer: We changed the believe to suggest.

Reviewer 3 Report

Although it is an interesting topic, the study is not clearly presented and needs improvement

  1. abstract should have the structure "Introduction-Methods-Results-conclusions"
  2. English language needs improvement
  3. Did they used questionnaires for the interview and they if they used they should mention if they were standarized
  4. I think it is better to analyze the answers and not just write the exact answers
  5. Discussion is to short compared with introduction and Methods
  6. In the discussion they do not mention other similar studies and they do no explain any differences 
  7. Another limitation is that they included only patients with communication skills. They do not explain what happens if the patients ca not communicate due to their illness
  8. They should put a flow chart of study selection
  9. The method is not clear enough. Authors searched the literature, excluded some studies and then they collected data from participants. They do not explain clearly the purpose of doing it., as this study is not a systematic review   

Author Response

1. abstract should have the structure "Introduction-Methods-Results-conclusions"

Answer: It was written more directly in line with the structure of the abstract.

2. English language needs improvement

Answer: Before submitting the revised manuscript, I received professional English editing.

  1. Did they used questionnaires for the interview and they if they used they should mention if they were standardized

Answer: We already wrote on Page 4 2) Data collect. Such as ;

The interview used semi-structured questions based on the attributes derived in the theoretical phase, and consisted of both specific and common questions. Common questions to both patients and nurses were: What do you think about safe medication? ; Tell me your role in the medication process; Tell us how to build the patient-nurse relationship for safe medication. The question specific to patients was: If there was a situation of partnership for your safety during injection or oral medication administration, please tell me about your feelings or experience at that time. The question specific to nurses was: If there was a situation in which you needed partnership with a patient in relation to medication care, please tell me about your experience or feelings at that time.

4. I think it is better to analyze the answers and not just write the exact answers

Answer: We add the sentences of analytic process on final analysis stage; In the final analysis stage, concepts and attributes that fit the purpose of the study were searched for while continuously comparing and analyzing the attributes of partner-ships derived from the theoretical and field work stages. As a result, the same properties derived in the theoretical and field research stages were maintained without modification. 

5. Discussion is to short compared with introduction and Methods

Answer: We add some sentences to describe the meaning of results.

6. In the discussion they do not mention other similar studies and they do no explain any differences

Answer: We add the comment on discussion; This results were similar with nurses involved the process of medication administration as an opportunity to engage patients in a conversation about medications and safety [13].

7. Another limitation is that they included only patients with communication skills. They do not explain what happens if the patients can not communicate due to their illness

Answer: We added the limitation according to reviewer’s comment.

8. They should put a flow chart of study selection

Answer: We add the PRISMA diagram with evidence of the article selection in <Figure 1>

9. The method is not clear enough. Authors searched the literature, excluded some studies and then they collected data from participants. They do not explain clearly the purpose of doing it., as this study is not a systematic review

Answer: We further described the hybrid model method in study methods. We also added the descriptions of each phase.

Round 2

Reviewer 3 Report

The authors answered to all comments. However I think it is better to use the structure Introduction-meterialias methods- results- conclusion for their abstract. It is easier for readers and to clear view of their study. 

Author Response

  1. The authors answered to all comments. However I think it is better to use the structure Introduction-meterialias methods- results- conclusion for their abstract. It is easier for readers and to clear view of their study.

We rewrite the abstract to according by reviewer’s opinion.

However, the subtitle of Introduction-method-result-conclusion is not specified in IJERPH's abstract writing regulations, so it is not divided separately.

Answer: Abstract: Medication safety is the most patient-centered aspect of nursing, and the medication process needs patients’ active participation to prevent medication errors effectively. The aim of this study was to develop the concept of a patient-nurse partnership for medication safety activities. The study design used the three-phase hybrid model for concept analysis and development was used: the theoretical phase, fieldwork phase, and final phase for integration. The results of a study define the concept of patient-nurse partnership for medication safety as “a fair cooperative relationship of mutual responsibility in which patients and nurses share information and communicate with each other through mutual trust.” Seven attributes were derived: mutual trust, mutual respect, mutual sharing, mutual communication, mutual responsibility, fair relationship, and mutual cooperation. Conclusion of results, in the patient-nurse partnership for medication safety, it is necessary to ensure a balance in power between patient and nurse. This balance can be established through patient-centered nursing by implementing the active transfer of authority from nurses as professionals to patients.
